# Laser Ranging Bathymetry Using a Photon-Number-Resolving Detector

**Lior Cohen** [1,†], **Daniel Istrati** [1,†], **Yoni Sher** [2,†], **Zev Brand** [3] **and Hagai S. Eisenberg** [1,*]

1   Racah Institute for Physics, Hebrew University of Jerusalem, Jerusalem 91904, Israel
2   School of Computer Science and Engineering, Hebrew University of Jerusalem, Jerusalem 91904, Israel
3   Independent Researcher, Modi'in 71700, Israel
*   Correspondence: hagai.eisenberg@mail.huji.ac.il
†   These authors contributed equally to this work.

**Abstract:** The sensitivity and robustness against background noise of optical measurements, and specifically range-finding, can be improved by detecting the light with photon-number-resolving detectors (PNRD). We use a PNRD to detect single pulse reflections from the seabed level in the presence of high attenuation of the sea water. Measurements are performed from above the sea level, overcoming broad daylight conditions. We demonstrate continuous measurement of the seabed depth up to around 24 m, using laser pulse energies of 10 μJ, while sailing at speed of 2.2 knots. Additionally, we use these data to extract values of the refractive index and optical attenuation in coastal seawater. The method could be used as a novel and optically-accurate bathymetry tool for coastal research and underwater sensing applications.

**Keywords:** laser rangefinding; seabed mapping; underwater optical sensing

## 1. Introduction

Laser ranging and imaging technologies have found many novel uses in the past few years, leading to accelerated development efforts [1]. One particular method—time-of-flight ranging—is based on sending and receiving a short laser pulse. By measuring the time of flight and knowing the speed of light in the medium the range is estimated:

$$r = 2t\frac{c}{n},\tag{1}$$

where $r$ is the range, $t$ is the measured time, $n$ is the index of refraction, and $c$ is the speed of light in vacuum. Autonomous navigation is a major use, as well as high resolution terrain mapping. Two interleaved goals for such systems are increasing the detected range [2] and using lower optical power. The ultimate physical sensitivity level is for single photons, which can be achieved by single-photon detectors [3,4] and photon-number-resolving detectors (PNRDs) [5,6].

Rangefinding and LIDAR systems are particularly limited in harsh environmental conditions with high scattering and low transmissivity, where underwater operation is a prominent example [7,8]. The large scattering and absorption of sea water severely hinders attempts to send an optical pulse and detect its reflection over useful distances. Specifically, these difficulties prevent the laser ranging and imaging of the seabed, either by a submerged system or by observations from above the sea level.

Seabed mapping has many applications, including oil and gas exploration, ocean science hydrographic surveying, marine archeology, and fisheries [9,10]. The leading method of seabed mapping is sonar, while sonar is accurate and reaches great depth [11], it has lower spatial resolution and is limited by the high reflection of the air-water interface, and thus, must be submerged [9]. The centimeters resolution of our device is comparable to sonar, but optical devices have an enormous improvement potential due to the much shorter

wavelength. The shorter wavelength enables, in principle, higher resolution [12] and faster scanning. Moreover, a laser system can provide a long-term solution to the increasing demand for higher and higher resolution [13]. We note that other optical methods are used for bathymetry, such as reported in Refs. [14,15], with some advantages but cannot compete with the depth resolution of time-of-flight ranging.

In this Letter, we present a single-photon sensitive water-penetrating laser ranging system. A photon-number detection is employed which enables high sensitivity as well as high dynamic range and background filtering. Thus, this detection provides a good solution to the scattering problem, because of its fast recovery from the blinding of the reflection from the sea surface. The system operates from above the water in broad daylight conditions. Thus, it could be suitable for operation from both watercraft and aircraft platforms, enabling a large area coverage with short scan times.

## 2. Methods

Our system has two optically and mechanically separated channels—a laser transmitter and a signal receiver (Figure 1a). The transmission channel is composed of a 100 mW Standa Nd-Yag micro-chip laser, passively mode locked and frequency doubled to 532 nm wavelength. This wavelength comprises a good trade-off between laser source complexity, water absorption and detector efficiency. The laser is triggered once for each ranging measurement, emitting a 0.5 ns, 80 µJ pulse with a maximum repetition rate of 1 kHz, theoretically limiting the range resolution to about 5 cm. The beam is expanded to a 3-cm diameter waist with a 300 mm-focus 2-inch-wide lens resulting in a divergence angle of about 12 microradians and Rayleigh length of about 5 km. The transmission channel also contains a polarizer used to attenuate the output power. For the experiments described here, the polarizer was tuned such that the system output was 10 µJ per pulse.

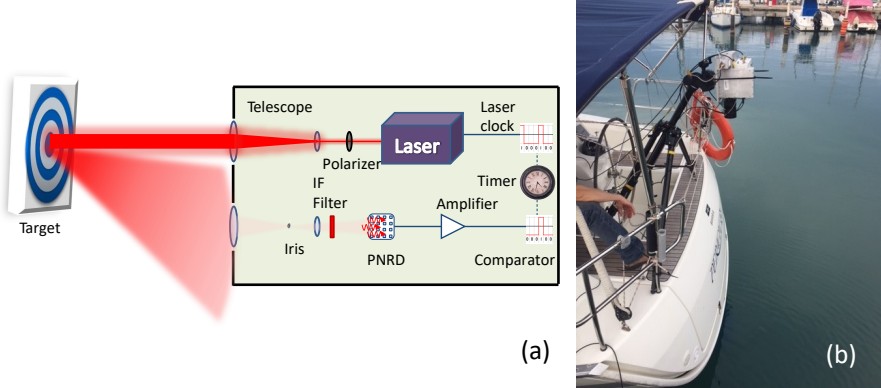

**Figure 1.** (**a**) Illustration of the rangefinder system. The transmission channel contains a beam expander (telescope) and a polarizer as detailed in Section 2. The receiving channel focuses the returning light onto the detector and uses a narrow filter to reduce background radiation (see also Ref. [6]). (**b**) The rangefinder, attached at the stern of the boat. Neither the boat nor the rangefinder used a stabilization protocol which reflects on the robustness of the device. The system is controlled by a laptop and data are acquired via a compact USB data acquisition card.

The receiving channel focuses the incoming light using a 175-mm-focus 3-inch lens with a field of view of about 7.5 milliradians. The light is detected with a PNRD, allowing for accurate reading of incoming light intensity down to the single-photon level [16]. The PNRD is a Hamamatsu Silicon photomultiplier (SiPM), composed of an array of 676 cells operated in Geiger mode, each capable of detecting a single photon at 532 nm wavelength with quantum efficiency of 40% [17]. The PNRD rise time is 1 nanosecond. A narrow bandpass filter of 3 nm centered around 532 nm reduces the background solar radiation reaching the detector. The signal from the PNRD is recorded by a compact high speed data acquisition card (DRS4 evaluation board from the Paul Scherrer Institut [18] with

1-nanosecond resolution) connected to and powered by a laptop. The laptop controlled data acquisition card allows for high system maneuverability and specifically setup in confined spaces with low available power sources such as boats or drones. The setup is about 30 cm × 30 cm × 10 cm in size and 4 kg in weight but can be significantly lightened up.

## 3. Results and Discussion

### 3.1. Refractive Index

Before the sea-water experiments, we tested the device in free space and measured ranges of more than 1 km, with about two orders of magnitude less power than regular rangefinders. We measured the single pulse depth resolution to be around 15 cm (which indicates ∼10-cm resolution in sea water).

For sea-water experiments, we first measure the refractive index of sea water from different distances from the shore. As we intend to present depth measurements from a travelling boat, where its distance from the shore is constantly changing, it is important to show that the time-of-flight of optical signals does not depend on the boat position through varying index of refraction of the water (see Equation (1)). Thus, we collected samples of sea water from three distances from the shore and brought them back to the lab. There, we filled a 1.8 m long glass water tank with the water samples whose sole purpose is to measure the index of refraction. Laser pulses were injected from one side of the tank, and reflected from a mirror inside the water at different distances. The time-of-flight was recorded for each mirror position. The refractive index is extracted from the measurement time dependence on the actual distance using Equation (1).

Figure 2 presents the collected data and linear fits. The fit slope is $\frac{c}{n}$, where $c$ is the speed of light in vacuum and $n$ is the refractive index. We measured values between $1.35 \pm 0.03$ and $1.365 \pm 0.03$ for water from different distances from the shore, while an independent measurement of the index of refraction from the same place is unavailable, empirical formulae produce a value of $1.340 \pm 0.005$ using the salinity and temperature at the actual experiment location [19], which is within good agreement to the experimental results. According to these results, the accuracy in the depth measurements is expected to be on the order of 1%.

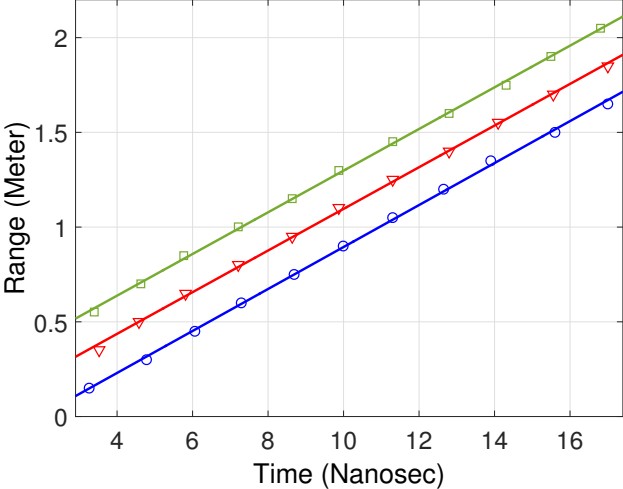

**Figure 2.** The pulse time-of-flight as a function of the range from the mirror, through seawater collected at different distances from shore and measured at the lab. Blue circles, red triangle and green squares represent data for seawater of 300, 1000 and 2400 m from shore, respectively. The measurements were displaced vertically by 0.2 m for presentation purpose only. The solid lines are linear fits to the data. The measured refractive indices are $1.35 \pm 0.03$, $1.365 \pm 0.03$, and $1.36 \pm 0.03$, respectively. All the values agree well, up to measurement error, with the measured values from previous works. Measurement errors are smaller than the symbol sizes.

### 3.2. Seabed Depth Ranging

The sea depth measurements were taken off a boat, sailing close to the Israeli shore of the Mediterranean sea. As described above, the pulse energy was set to 10 µJ. The main challenge is that the reflected signal from the sea surface saturates the amplified signal from the detector, while the reflected signal from the seabed is much weaker. In order to overcome this problem, we periodically changed the bias voltage over the SiPM through five values each second (see also Appendix A and Figure A1 there). Higher bias voltages result in higher sensitivity but more blinding, enabling detection of deeper objects. In the presented results the blind time was up to about 50 ns (which is also the dead time of a single SiPM cell). Lower bias voltages result in lower sensitivity but less blinding, enabling detection of shallower objects. The signal was acquired by a USB acquisition card operated and powered directly from a laptop, allowing real-time display and analysis of the recorded data. After numerically differentiating the recorded signal, target detection was marked by surpassing a predefined positive threshold.

First, we measured the depth at several points and found a good match with the on-board sonar (all measurements within 20 cm agreement). A typical result is shown in Figure 3a. The blue line is the raw voltage produced by the PNRD. As expected, the signal returned from the water's surface saturates the detector, while the seabed signal is much weaker as a result of the optical attenuation in the water. The points of detected reflections are marked by red dots. The signal returned from the surface also causes a deleterious effect of blind ranges: since high sensitivity is required to find the seabed, the high intensity returned from the sea-level surface temporarily blinds the detector. To overcome this temporal blinding, we cluster the detected points into a single point, removing the multiple surface detection that can be seen in Figure 3a. The real-time analysis correctly detects the water's surface, which moves up and down with the waves (see Figure 3c), and the seabed. While the range to the sea level is changed by the amplitude of the waves, the range to the seabed is fixed (see Figure 3d). The time of flight to the seabed is changed a bit because the light travels more distance in water for wave maxima. Thus, in the measurement of the sea bottom we expect about one fourth of the change of the wave amplitude which is caused by the difference in index of refraction. This can be taken into consideration when calculating the seabed depth. Additional detection points are occasionally seen in between, resulting from reflection from the water's bulk or an occasional fish.

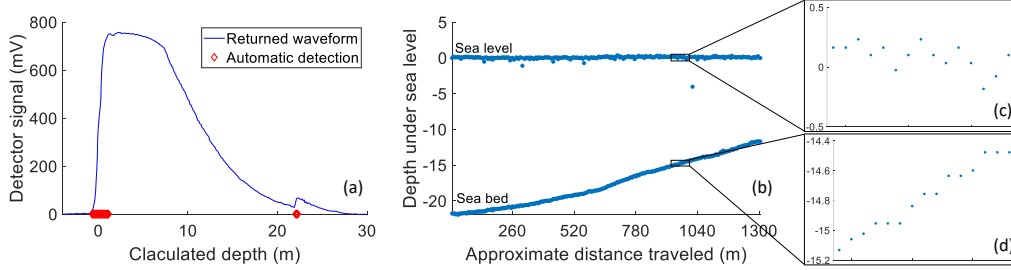

**Figure 3.** (**a**) Raw waveform data as measured by the SiPM detector, with detected object ranges. (**b**) Detected return signals from the sea's surface and the seabed. The signal from the surface is highly variable due to waves, while the seabed's depth changes slowly as the boat sail toward the shore. (**c**) Zoom-in example of the sea level which reveals waves of up to 40 cm change in sea level. (**d**) Zoom-in example of the seabed signal, corresponding to the sea level signal in (**c**). No correlations are seen between the two signals.

After the seabed ranging had been established, we operated the rangefinder while in motion and continuously tracked the sea-bed level at a rate of about 0.5 Hz. Additional software and hardware adjustments can increase the detection rate up to the 1 kHz repetition rate of the laser. However, for detecting the seabed, 0.5 Hz is a good balance. Figure 3b shows the processed data as a function of the distance travelled between a starting position

which is about 2000 m from the shore to about 700 m from the shore. The depth was calculated considering a constant refractive index of 1.35. Clear and continuous measurements of the sea depth were achieved.

### 3.3. Optical Attenuation in Seawater

The attenuation of the optical pulse when propagating through seawater is a result of both absorption and scattering. Although the used wavelength is close to the minimal absorption point, the effect of both causes is comparable [20]. The raw signal can be used to evaluate the returned relative intensity. Assuming a uniform albedo of the seabed, this measurement can be used to extract the sea water attenuation of the transmitted light. Choosing a single detector bias voltage of the 5 values that were used, at depths smaller than 17 m the returned signal either saturates the detector or is close to saturation and therefore nonlinear. The equation describing sensor saturation is $\bar{n}_s = N(1 - e^{-\frac{\bar{n}}{N}})$, where $N$ is the number of sensor elements and $\bar{n}$ is the average number of incident photons [21]. We assume exponential attenuation at depth $x$, $\bar{n} \propto I = I_0 e^{-cx}$. Thus, by combining the two equations, the normalized sensor activation in decibel (dB) units as a function of the depth is:

$$I_s = 10 \cdot log_{10}\left(1 - exp\left(-\frac{1}{N}exp(-c*x+d)\right)\right).$$

$N$ is the number of elements in the sensor, $c$ is attenuation per meter in dB, and $d$ is an offset parameter, found by a linear fit to the results.

In Figure 4, the relative attenuation, calculated from the raw data of Figure 3a, is plotted as a function of the seabed depth. The data were fit to the attenuation detailed above. The attenuation coefficient of sea water at these depths was found to be $2.3 \pm 0.1$ dB/m, with $N = 663$, a close match to the number of elements in the sensor (676).

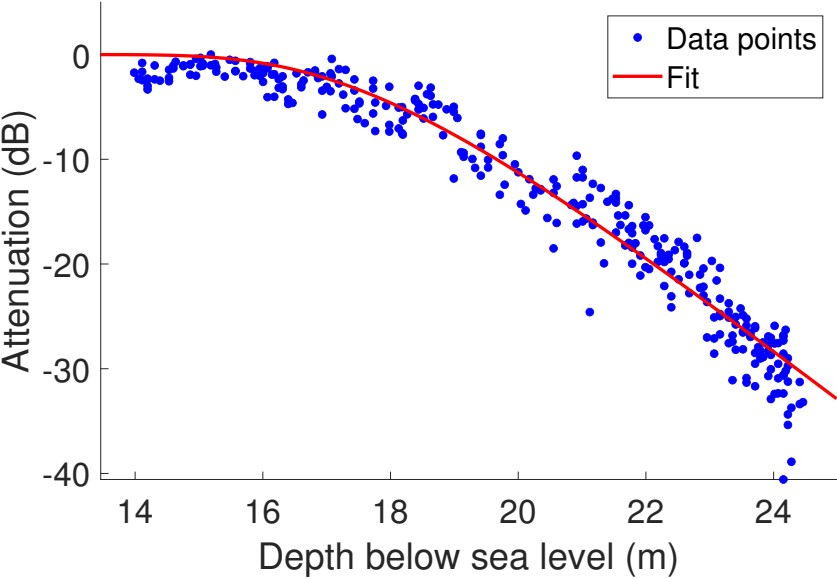

**Figure 4.** Attenuation of the returned signal as a function of depth. The fitted value is $4.6 \pm 0.2$ dB/m for the round trip, implying a value of $2.3 \pm 0.1$ dB/m attenuation for a single pass. The fit's correlation coefficient is $R = 0.96$.

The setup can be modified based on application requirements. For example, if the depth is limited to be greater than a few meters, fewer bias voltages are needed which will result in faster measurements. If a higher range is required, a stronger laser by a factor of 4.6 dB per meter is needed (and see also Ref. [9] for laser powers used in conventional LIDARs). Signal processing methods, such as the one reported in Ref. [22], can improve the target contrast and are compatible with our device.

## 4. Conclusions

We have demonstrated a system for measuring the depth of shallow water from above the surface. The results show continuous measurements of the seabed for depths of 10–20 m while sailing at a speed of 2.2 knots. Moreover, the dynamic range of the system further allows for seabed detection at depths either smaller than a meter or larger up to 24 m. As an application, sea water optical attenuation coefficient of 2.3 dB per meter has been measured. The result is within the literature values for intensity attenuation, which are prone to location variability. Tests conducted on dry land indicate that this system could operate from a height of several hundred meters with similar results. Thus, a more compact and rugged version could be used with a small aircraft or drone for seabed mapping and depth measurements in shallow areas such as coastal waters and rivers. The presented method increases the sensitivity and the dynamic range of laser rangefinders. These parameters are in particular important in underwater ranging because of the high attenuation. Thus, this method may give new capabilities for coastal research and underwater sensing applications.

**Supplementary Materials:** The following supporting information can be downloaded at: https://www.mdpi.com/article/10.3390/rs14194750/s1.

**Author Contributions:** Conceptualization, H.S.E.; methodology, L.C., D.I., Y.S. and H.S.E. software, Y.S.; formal analysis, L.C., D.I., Y.S. and H.S.E.; investigation, L.C., D.I., Y.S. and Z.B.; resources, Z.B.; data curation, L.C., D.I., Y.S. and Z.B.; writing—original draft preparation, L.C., D.I. and Y.S.; writing—review and editing, L.C., D.I., Y.S. and H.S.E.; visualization, L.C., D.I., Y.S. and H.S.E.; supervision, H.S.E.; project administration, Z.B. and H.S.E.; funding acquisition, Z.B. and H.S.E. All authors have read and agreed to the published version of the manuscript.

**Funding:** This research received no external funding.

**Data Availability Statement:** Raw data and codes, used to generate the figures can be found in the supplementary material and are available as ancillary Matlab files with the arXiv posting of this Letter.

**Conflicts of Interest:** The authors declare no conflict of interest.

## Abbreviations

The following abbreviations are used in this manuscript:

| | |
|---|---|
| LIDAR | light detection and ranging |
| PRND | photon-number-resolving detector |
| SiPM | silicon photo-multiplier |
| USB | universal serial bus |
| nm | nanometers |
| ns | nanoseconds |

## Appendix A

The number of 5 bias voltages was chosen to ensure the dynamic range of depth. Smaller number of voltages would degrade the dynamic range while a higher number would slow the measurement. In Figure A1, an example of the five voltages is shown. The seabed is detected at 22.5 m for the two higher voltages. Then, If the seabed is detected for different bias voltages in a time window of 2.8 ns, the median was chosen as the displayed detected range. The group of automatic detections near the sea level is a result of reflections from multiple water layers. This is expected from water scattering.

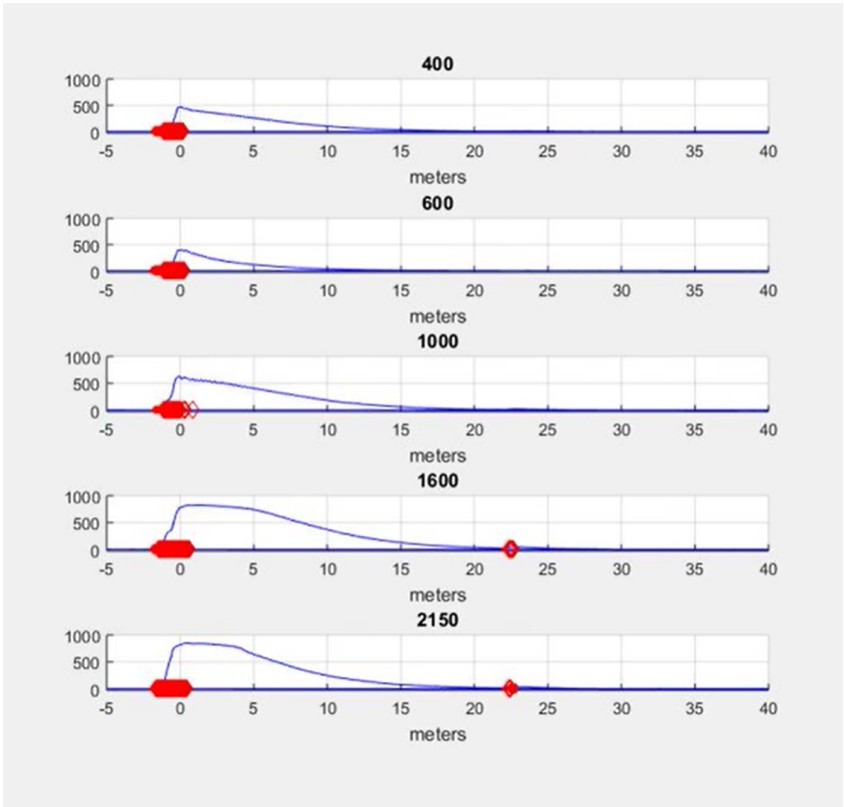

**Figure A1.** Single cluster of five voltages. Voltage is displayed in arbitrary units. 2150 is the highest voltage used, beyond that the detector is unstable. The quantum detection efficiency is about 40% for voltage of 2150 and gets lower as the voltage is decreased. The x-axis is calibrated to show the range based on Equation (1) and the measurement of the refraction index in Figure 2.

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
