# Peer review of "Laser Ranging Bathymetry Using a Photon-Number-Resolving Detector"

_remotesensing, doi:10.3390/rs14194750_

Round 1
Reviewer 1 Report (Previous Reviewer 6)
The authors adequately addressed the reviewers' comments.
Author Response
We thank the reviewer for the second review.
Reviewer 2 Report (Previous Reviewer 5)
Through revising , the contain was enough for publishing from reviewer's point of view. Conclusion part can be more meaningful and thoroughly summary about merits. That is suggestion for author's thinking.
Author Response
We thank the reviewer for the second review. We agree that the advantage of the method should be stated more clearly in the conclusion section. We reorganized this section and added the bolded text:
“We have demonstrated a system for measuring the depth of shallow water from above the surface. The results show continuous measurements of the seabed for depths of 10-20 m while sailing at a speed of 2.2 knots. Moreover, the dynamic range of the system further allows for seabed detection at depths either smaller than a meter or larger up to 24 m. As an application, sea water optical attenuation coefficient of 2.3 dB per meter has been measured. The result is within the literature values for intensity attenuation, which are prone to location variability. Tests conducted on dry land indicate that this system could operate from a height of several hundred meters with similar results. Thus, a more compact and rugged version could be used with a small aircraft or drone for seabed mapping and depth measurements in shallow areas such as coastal waters and rivers. The presented method increases the sensitivity and the dynamic range of laser rangefinders. These parameters are in particular important in underwater ranging because of the high attenuation. Thus, this method may give new capabilities for coastal research and underwater sensing applications.”
Reviewer 3 Report (Previous Reviewer 2)
I thank the authors for their careful and comprehensive response to the review questions and comments.
Author Response
We thank the reviewer for the second review.
This manuscript is a resubmission of an earlier submission. The following is a list of the peer review reports and author responses from that submission.
Round 1
Reviewer 1 Report
Summary
This study introduced a laser ranging bathymetry instrument and some results using a Photon-Number-Resolving Detector.
However, the instrument about Photon-Number-Resolving Detector has not adequately described, signal comparisons between PNRD’s performance and common PMT should need be presented. The results have not yet clearly presented, which is to easy and just a experiment. I suggest the authors carried out some more field experiments and obtained more data, then to submit again.
Reviewer 2 Report
This manuscript describes a series of underwater lidar experiments for measuring water depth from a moving surface platform. The lidar instrument is mostly adequately described, and the ranging experimental data are presented clearly. The experimental design is very well done, especially the laboratory measurements of the index of refraction. Several elements could be improved prior to publication, however. In general these include a longer and more in-depth discussion of the state of the art of underwater lidar (including further relevant references), a better discussion of the applicability of these results to future instrument design and the technique as a whole, and better presentation of a comparison between the sonar and lidar data to show the sea floor ranges were accurate. Specific points of improvement are given below, and one these are addressed I would consider the manuscript fit for publication.
· The introduction briefly touches on the topic of using laser-based methods of seabed mapping, but the work presented here would be well served by more detail about the benefits that such a technique could provide over sonar. The only reference given (reference 11) is over 30 years old, so more recent work should be cited and properly contextualized. For example, the 2013 review of oceanographic lidar by Churside would be a key reference.
· Micropulse and pulse coding techniques have been recently applied to underwater ranging, and should be discussed in comparison with the single pulse technique demonstrated here (ex. Cochenour et al, 2011, Applied Optics)
· The caption of Figure 1 has a section number that needs to be updated
· Please state the transmitted beam divergence and receiver field of view
· What was the timing resolution of the data acquisition system? What was the bandwidth or rise time of the SiPM? These could both affect the ranging resolution and should be given.
· Why was the beam attenuated to 10 microjoules per pulse? If this was to avoid saturation from the sea surface signal please state that explicitly.
· What was the bandwidth of the narrow-band filter?
· Figure 2 is somewhat misleading based on the authors’ decision to offset the curves. My opinion is that if the goal is to demonstrate the similarity of the refractive indices from the three locations (and the linearity), that the three symbols can be plotted without any offset and without the linear fits. The linearity of the data are obvious without the linear fits shown.
· What is the footprint of the sonar measurement on the seafloor and how does it compare to the lidar? Could slopes or seabed topography cause differences in the measurements?
· How long is the “blind time” of the detector?
· What were the sea surface conditions during the experiments? Does the rise and fall of the boat due to waves affect the sea bed range measurements or is it minimal?
· When continuously changing the bias voltage of the SiPM five times a second, how are the ranges obtained at 0.5 Hz? Is the best data from those 10 different bias measurements taken or are the data averaged over that entire time? Perhaps a figure showing 2 seconds of raw data and how the data are reduced would be helpful here.
· Line 96: How is this clustering done? Is an average of the detected points taken? Is the first detection point used?
· Figure 3 states the signal from the surface is highly variable but it looks relatively flat here. Has the data in Figure 3b been normalized by the sea level value? Perhaps an additional figure frame that shows a smaller section of distance (10 or 20 m) will show the wave effect better.
· Figure 3b would be improved by also including the sonar range profile in a different color to show how it compares to the seabed.
· How does your measured value of the attenuation coefficient compare to previously measured values for the water you measured in? Please provide numerical values and references.
· What do your results indicate about the use of this technique more widely? It appears as if the surface reflection and water attenuation problems require very high dynamic ranges to accommodate. If a system required a higher range of depth capabilities (say 5 to 50 m?) what would that mean for the system design?
· The conclusion states the instrument is capable of operating on a drone. What is the current mass and power of the system, and what are the capabilities for reduction in a future instrument?
Reviewer 3 Report
This letter reported the experiment result of laser ranging of seabed using a lidar with photon-number-resolving detector. The main advantage of the method/instrument is the improvement of sensitivity. There are significant improvement needed:
1. There is no details about the analysis/evaluation of the lidar system performances, the issues (such as the signal and noise characteristics, the waveform characteristics, the range estimation methods, ... etc) to be solved for the system.
2. There is no analysis of the properties for the sea water, as well as the attenuation, scattering characteristics, which can support the processing analysis.
3. Innovative research is not addressed, especially there is no comparison with other available technology for this application.
Reviewer 4 Report
This paper uses a PNRD to detect single pulse reflections from the seabed to detect sea bathymetry, and the system is evaluated by experiments. The structure of the paper is well organized. However, the principles and results need to be clearly stated. Specific comments are as follows.
The method to estimate bathymetry by the system should be given in detail, such as the key formulas, variables and parameters, etc.
The results are not evaluated by in-situ measurements, e.g., the bathymetry measured by ADCP.
Is there any comparison with traditional Lidar? If it is available, the advantages of the system may be distinct.
Figure 1: “as detailed in section ??”
Reviewer 5 Report
1. For the fig3a, the procedure on process and get automatic detection results , as shown as red points , should be more clear , and for now , it was not easy for understanding. Such as first group red points and the second group red point , what meaning for these ?
2. As context between line 64 and line 66, ‘There, we filled a 1.8 m long glass water tank with the water samples, injected the range-finder laser pulses from one side of the tank, and reflected them from a mirror inside the water at different distances’ , Sea floor is quite different from the regular tank, from reflection ability and roughness etc
3. For the conclusion, the speed motion , data rate and limit of maxim and minimal range for detector ,should be analysis.
Reviewer 6 Report
Lines 78 and 90: The problem of sensor saturation due to reflection from water surface is discussed twice in almost the same words.
Figure 3: Was the rangefinder rigidly attached to the boat? Was the boat somehow stable with respect to the geoid?
Authors did not explain why the measured distance to water surface is constantly changing due to waves while waves are not affecting seabottom measurements.
The authors propose to use a single channel green laser rangefinder for bathymetric measurement. In my opinion, this is a simplified version of a time-of-flight camera, which has been demonstrated in the paper by K.Mack, W. Jemison, L. Rumbaugh, D. Illig, M. Banavar "Time-of-Flight (ToF) Cameras for Underwater Situational Awareness" presented at Oceans 2019.